# Arabidopsis RNA processing factor SERRATE regulates the transcription of intronless genes

Corinna Speth[1,2,3], Emese Xochitl Szabo[1,2,3,4], Claudia Martinho[1,2,3], Silvio Collani[5], Sven zur Oven-Krockhaus[1], Sandra Richter[1], Irina Droste-Borel[6], Boris Macek[6], York-Dieter Stierhof[1], Markus Schmid[5], Chang Liu[1], Sascha Laubinger[1,2,3,4]*

[1]Centre for Plant Molecular Biology (ZMBP), University of Tuebingen, Tuebingen, Germany; [2]Chemical Genomics Centre (CGC) of the Max Planck Society, Dortmund, Germany; [3]Max Planck Institute for Developmental Biology, Tuebingen, Germany; [4]Institute for Biology and Environmental Science, University of Oldenburg, Oldenburg, Germany; [5]Department of Plant Physiology, Umea Plant Science Centre, Umeå University, Umea, Sweden; [6]Proteome Centre, University of Tuebingen, Tuebingen, Germany

*For correspondence:
sascha.laubinger@uol.de

Competing interests: The authors declare that no competing interests exist.

**Abstract** Intron splicing increases proteome complexity, promotes RNA stability, and enhances transcription. However, introns and the concomitant need for splicing extend the time required for gene expression and can cause an undesirable delay in the activation of genes. Here, we show that the plant microRNA processing factor SERRATE (SE) plays an unexpected and pivotal role in the regulation of intronless genes. Arabidopsis SE associated with more than 1000, mainly intronless, genes in a transcription-dependent manner. Chromatin-bound SE liaised with paused and elongating polymerase II complexes and promoted their association with intronless target genes. Our results indicate that stress-responsive genes contain no or few introns, which negatively affects their expression strength, but that some genes circumvent this limitation via a novel SE-dependent transcriptional activation mechanism. Transcriptome analysis of a Drosophila mutant defective in ARS2, the metazoan homologue of SE, suggests that SE/ARS2 function in regulating intronless genes might be conserved across kingdoms.
DOI: https://doi.org/10.7554/eLife.37078.001

## Introduction

Regulation of gene expression is fundamental for all aspects of eukaryotic life. In plants, developmentally or stress-induced changes in gene expression are essential for plant growth, development and defense. Gene expression can be controlled at various levels, including transcription, RNA processing or translation. Accurate gene expression at the level of RNA processing includes various steps, including the attachment of a 7-methylguanosine ($m^7G$) cap to the 5' end (capping) of the nascent mRNA, followed by intron excision, exon ligation (splicing), and 3'-end formation via transcript cleavage and polyadenylation. Transcription and RNA processing occur simultaneously and are mechanistically coupled, but how specific RNA processing factors influence transcription is not fully understood (*Bentley, 2014*).

Arabidopsis (*Arabidopsis thaliana*) SERRATE (SE) is an essential, conserved eukaryotic RNA processing factor important for plant development (*Clarke et al., 1999*; *Prigge and Wagner, 2001*; *Grigg et al., 2005*; *Lobbes et al., 2006*; *Wilson et al., 2008*). SE null alleles are lethal, while hypermorphic alleles such se-1 or se-3, which carry small deletion or T-DNA insertions, respectively, display a wide range of developmental abnormalities (*Clarke et al., 1999*; *Grigg et al., 2005*;

*Lobbes et al., 2006*). The SERRATE protein possesses distinct domains that mediate protein-protein interactions and binding to GGN repeats in RNAs (*Machida et al., 2011*; *Iwata et al., 2013*; *Foley et al., 2017*). SE is probably best known for its function in the microRNA (miRNA) pathway (*Grigg et al., 2005*; *Lobbes et al., 2006*; *Yang et al., 2006*). SE and its metazoan ortholog ARSE-NITE RESISTANCE2 (ARS2) form complexes with DICER proteins and are required for efficient, precise primary-miRNA processing (*Sabin et al., 2009*). Furthermore, SE/ARS2 participates in other RNA maturation steps, including constitutive and alternative splicing of mRNAs, 3'-end formation, biogenesis of non-coding RNAs (ncRNAs), RNA transport and RNA stability (*Laubinger et al., 2008*; *Laubinger et al., 2010*; *Gruber et al., 2012*; *Hallais et al., 2013*; *Raczynska et al., 2014*). ARS2 also activates the transcription of *SOX2*, a gene involved in stem cell maintenance (*Andreu-Agullo et al., 2011*). SE/ARS2 executes its function in conjunction with the nuclear cap-binding complex (CBC), which consists of two subunits (CBP20 and CBP80) and binds to $m^7G$-caps at the 5' ends of polymerase II (pol II)-derived transcripts (*Laubinger et al., 2008*; *Hallais et al., 2013*; *Raczynska et al., 2014*). Many RNA processing events involving SE occur co-transcriptionally (*Fang et al., 2015*), but whether SE plays important roles in coupling RNA processing with transcription is currently unknown. In this study, we examined the function of SE during transcription and found that SE is a regulator of intronless genes.

## Results and discussion

### SERRATE associates with the chromatin of intronless genes

We reasoned that identifying SE binding sites in the Arabidopsis genome might help us elucidate the role of SE in co-transcriptional gene regulation and uncover novel functions for SE. We therefore carried out chromatin immunoprecipitation (ChIP) experiments using a SE-specific antibody in WT and a hypomorphic se mutant (*se-1*) as a negative control, followed by sequencing (ChIP-seq). We identified 1012 high-confidence SE binding sites ('peaks') in three independent biological replicates that were enriched in WT when compared to *se-1* (*Figure 1A,B*, all peaks are listed in *Supplementary file 1*). To confirm the ChIP-seq results, we selected 12 SE target loci and assessed their association with SE by ChIP-qPCR. All 12 loci showed enrichment for SE in WT, but not in se-*1* (*Figure 1C*). Thus, our experiments revealed that SE directly associates to specific regions in the Arabidopsis genome.

We found that SE peaks were primarily located in exonic regions and/or close to transcriptional start sites (TSSs) (*Figure 1D*). The SE peaks were often very broad and covered entire genes or large portions of genes (*Figure 1B*). These observations suggest that the association of SE with chromatin depends on transcription. A search for motifs enriched among SE targets revealed that, among others, a SE-binding GGN repeat is enriched among the SE target genes (*Supplementary file 2*). Blocking RNA production by treatment with the transcriptional inhibitor cordycepin reduced the association of SE with all target genes examined, suggesting that association of SE with chromatin is indeed RNA-dependent (*Figure 1E*). We also found that mutations in a gene encoding a CBC component, *CBP20*, completely blocked the association of SE to chromatin (*Figure 1F*). ChIP-qPCR with CBP20-specific antibodies showed that also CBP20 associated with all SE targets examined; this association was attenuated in the se mutant (*Figure 1G*). All of these results suggest that the binding of SE to its target genes depends on RNAs and the CBC.

The observations that SE mainly associates with gene bodies and requires transcription for its association with chromatin raised the possibility that SE associates with chromatin during co-transcriptional RNA processing of pri-miRNAs or splicing of mRNAs. However, of 325 Arabidopsis *MIRNA* genes, only nine were bound by SE (p=0.8, hypergeometric test) (*Figure 1—figure supplement 1*). Therefore, our ChIP-seq data do not provide evidence for a widespread effect of co-transcriptional *MIRNA* processing. Instead, SE mainly associated with protein-coding gene loci (*Figure 1—figure supplement 1*). Since SE regulates splicing, we asked whether SE target genes are particularly intron-rich. Unexpectedly, 46.3% of the SE peaks were associated with intronless genes (ILGs, *Figure 1H*). In general, SE target genes possessed significantly fewer introns than average Arabidopsis genes (*Figure 1I*). Only 10.2% of the SE target genes contained more than five introns (*Figure 1H*). These results indicate that SE preferentially associates with ILGs, suggesting that SE plays a novel role at these loci.

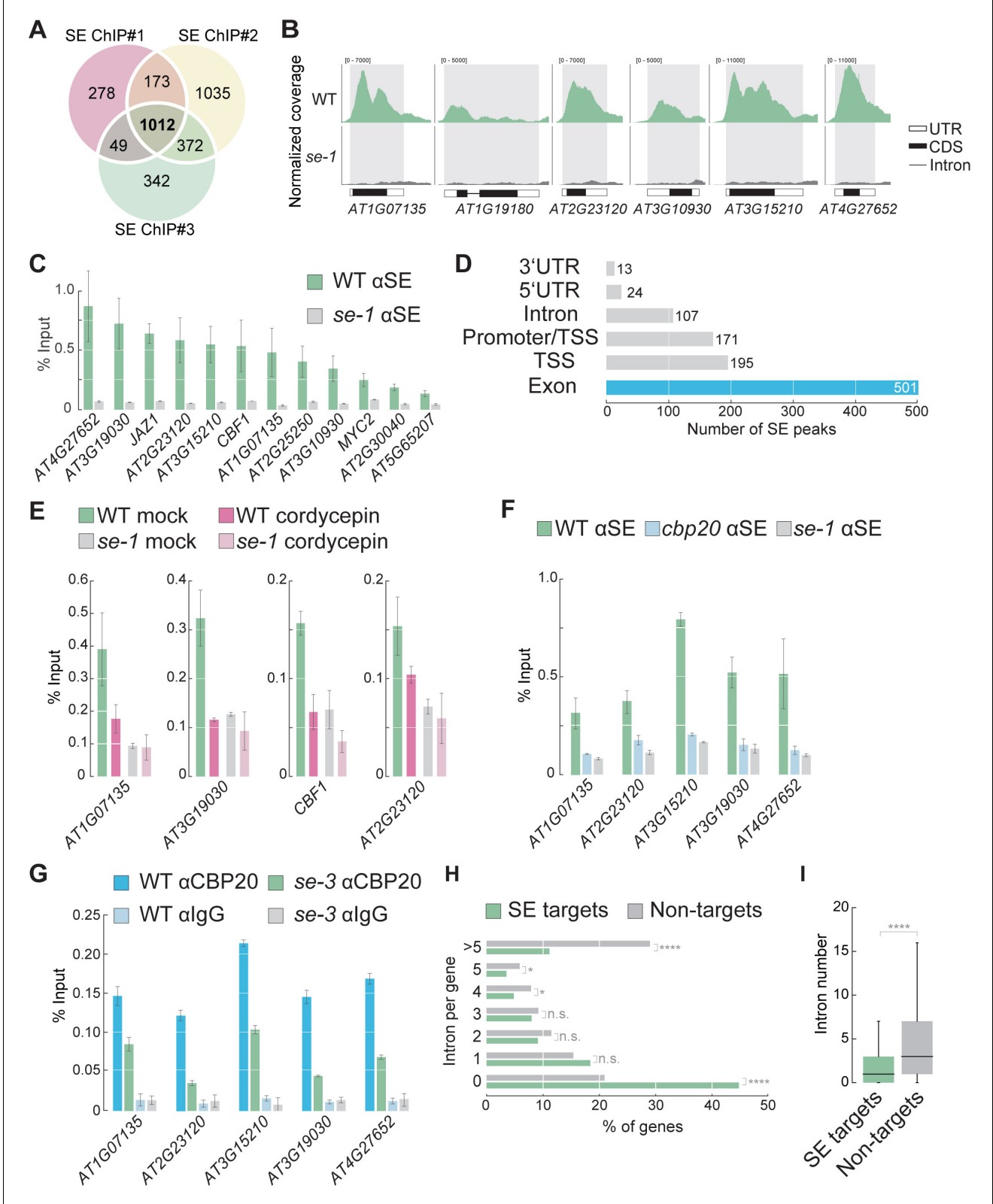

**Figure 1.** SE associates with intronless genes in a transcription dependent manner. (A) Venn diagram showing the overlap of SE ChIP-seq targets in three independent biological replicates. (B) Visualization of SE ChIP-seq data in WT and *se-1*. Tracks showing counts of sequencing reads mapped to the depicted genomic loci. (C) Validation of SE targets by ChIP-qPCR using SE-specific antibodies in WT and *se-1* mutants. Quantification of enriched DNA fragments was performed by qPCR. Error bars indicate the range of two independent biological experiments. (D) Annotation of the 1012 SE-ChIP

*Figure 1 continued on next page*

*Figure 1 continued*

targets sites. Peaks are categorized in six distinct classes: promoter-transcription start site (promoter-TSS), transcription start site (TSS), 5'-UTR, exon, intron, 3'-UTR. Y-axis denote the number SE peaks within each category. (E) Analysis of SE enrichment at selected targets by ChIP-qPCR in the presence and absence of the transcriptional inhibitor cordycepin. Error bars indicate mean ± SEM of three independent biological replicates. (F) Analysis of SE enrichment at SE target loci in WT, *se-1* and *cbp20* mutants by ChIP-qPCR. Error bars indicate mean ± SEM of three independent biological replicates. (G) Analysis of CBP20 enrichment at SE target loci in WT and *se-3* mutants by ChIP-qPCR using a CBP20-specific antibody. Rabbit IgG served as a background control. Error bars indicate mean ± SEM of three biological replicates. (H) Classification of SE target genes and non-targets based on intron number. A Fisher's exact test was performed to access whether differences between SE targets and non-targets were significant. *$p < 0.05$; ****$p < 0.0001$; n.s, not significant (I) Box blots comparing number of introns per gene in SE target genes and non-targets. SE targets are significantly enriched for low numbers of introns compared to non-SE targets (Wilcoxon-Mann-Whitney test). ****$p < 0.0001$.

DOI: https://doi.org/10.7554/eLife.37078.002

The following figure supplement is available for figure 1:

**Figure supplement 1.** SE mainly binds to protein-coding genes.

DOI: https://doi.org/10.7554/eLife.37078.003

## Intronless genes are expressed at low levels, but SERRATE enhances expression of a subset of intronless genes

Since SE bound preferentially to ILGs, we investigated the characteristics of intronless versus intron-rich genes. Intron splicing offers opportunities for the regulation of gene expression at various levels; introns positively influence transcription, RNA stability, and translation in a process known as 'intron-mediated enhancement of gene expression' (*Laxa, 2016*). In agreement with the finding that introns positively affect gene expression, we found that Arabidopsis genes with no or few introns exhibited lower maximum expression levels across a wide range of developmental stages and stress responses compared to intron-rich genes (*Figure 2—figure supplement 1*). It is possible that introns are detrimental when changes in gene expression must occur rapidly, since the transcription of introns and pre-mRNA splicing are time-consuming processes that increase the time needed for a gene to be expressed and reduce the responsiveness of the gene to various conditions (*Jeffares et al., 2008*; *Swinburne and Silver, 2008*; *Zhu et al., 2016*). Due to their sessile lifestyle, plants must adapt to a plethora of environmental changes very quickly, which likely explains the relatively high abundance of ILGs in plant genomes compared to other higher eukaryotes (*Jain et al., 2008*; *Wu et al., 2013*). Indeed, our analysis of changes in gene expression in response to drought, heat, cold, or salt in Arabidopsis and major crops such as rice, maize, and soybean showed that genes whose expression changes in response to stress contain fewer introns than average genes (*Figure 2—figure supplement 2*). These results indicate that genes lacking or with very few introns are important components of the plant stress transcriptomes of monocot and dicot plant species irrespective of the stress applied, and they predict that there is a major trade-off between gene expression strength and gene responsiveness during stress.

Because SE mainly associates with many ILGs, we asked whether SE influences their expression or stress-dependent regulation. In general, genes that physically associated with SE exhibited significantly higher expression levels than genes not bound by SE (*Figure 2A*). ILGs bound by SE were expressed at significantly higher levels than non-SE targets (*Figure 2B,C*). The expression of SE targets containing one or two introns was even higher than that of ILGs bound by SE, suggesting that introns can further enhance the expression of SE target genes (*Figure 2B,C*). Histone marks indicative of active transcription (such as acetylation of histone H3K9, H3K18, and H3K23) were enriched among SE targets compared to non-targets, suggesting that SE target genes are highly transcriptionally active (*Figure 2D–F*). Indeed, global analysis of RNA polymerase II (pol II) occupancy showed that SE target genes were more highly associated with pol II than are non-SE targets (*Figure 2G*). These results suggest that the strong expression of SE target genes, most of which are ILGs, is achieved at the transcriptional level. To investigate whether the presence of functional SE protein is necessary to maintain the high expression levels of its targets, we generated the *se-1* mutant transcriptome via mRNA sequencing (RNA-seq). A significant proportion of direct SE-target genes were downregulated in *se-1* (*Figure 2H*), suggesting that the presence of functional SE is indeed necessary to maintain the high expression levels of some of its target genes. Because *SE* is an essential gene, we could not investigate the expression of SE target genes in a *se*-null mutant background. However, we observed consistent downregulation of several selected intronless SE target genes in

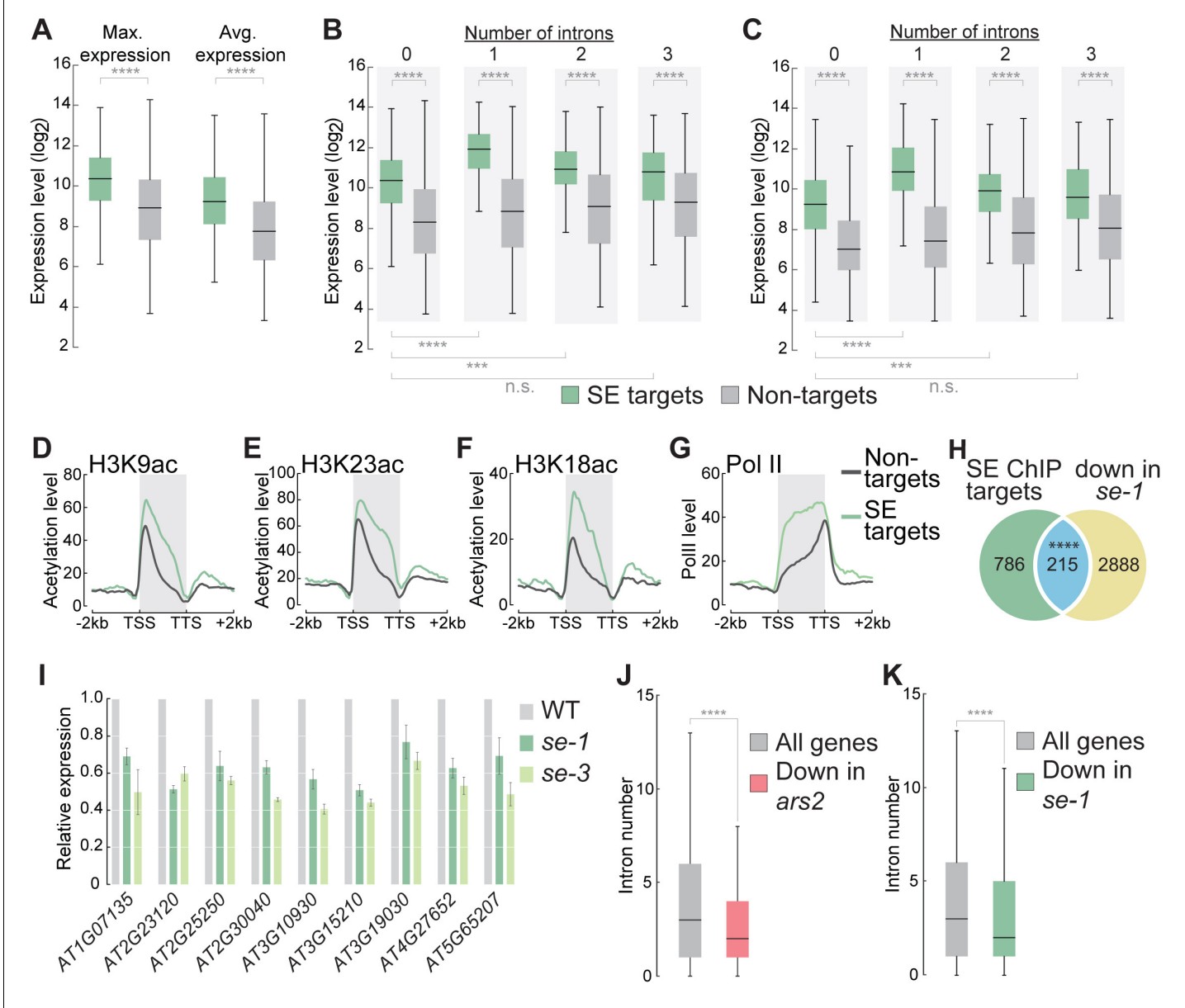

**Figure 2.** SE association to intronless genes indicates and maintains high gene expression levels. (**A**) Box plots showing the maximum and average gene expression of SE targets and non-targets across different Arabidopsis tissues and developmental stages (described in *Laubinger et al., 2008*). (Wilcoxon-Mann-Whitney test, ****p<0.0001) (**B,C**) Box plots showing the maximum (**B**) and average (**C**) gene expression of intronless and intron-containing SE targets and non-targets across different Arabidopsis tissues and developmental stages (described in *Laubinger et al., 2008*). (Wilcoxon-Mann-Whitney test, ***p<0.001; ****p<0.0001; n.s., not significant) (**D–G**) Profiles of histone H3K9, (**B**) H3K18 (**C**) and H3K23 (**D**) acetylation and pol II (**D**) levels over SE target genes and non-target genes. (**H**) Venn diagram showing the overlap between SE-ChIP targets and genes that are significantly down-regulated in *se-1* mutants (hypergeometric test, ****p<0.0001). (**I**) Quantification of indicated RNA transcript level determined by qPCR in WT, *se-1* and *se-3* mutants. Error bars indicate mean ± SEM of three biological replicates. (**J,K**) Box plots depicting the number introns of all *Drosophila melanogaster* (**J**) and *Arabidopsis thaliana* (**K**) genes and genes whose expression is significantly down-regulated (to < 50%) in *ars2* (**J**) and *se-1* (**K**) mutants. (Wilcoxon-Mann-Whitney test, ****p<0.0001).

DOI: https://doi.org/10.7554/eLife.37078.004

The following figure supplements are available for figure 2:

**Figure supplement 1.** Arabidopsis genes containing introns are expressed at higher levels.
DOI: https://doi.org/10.7554/eLife.37078.005

**Figure supplement 2.** Stress-regulated genes contain fewer introns in diverse plant species.
DOI: https://doi.org/10.7554/eLife.37078.006

*Figure 2 continued on next page*

*Figure 2 continued*

**Figure supplement 3.** SE binds and affects the expression strength of *CBF* genes.

DOI: https://doi.org/10.7554/eLife.37078.007

another hypomorphic *se* mutant (*se-3*) by RT-qPCR, further confirming that SE is necessary to maintain the high expression levels of its intronless target genes (*Figure 2I*). Some of the genes examined are well-established marker genes for stress-induced gene expression, including the intronless and cold-responsive *CBF* genes (*Figure 2—figure supplement 3*) (*Thomashow, 1999*). To investigate whether the induction of these genes under stress conditions also depends on SE, we exposed WT and *se-1* plants to cold stress for 15 and 60 min and analyzed the expression of SE-bound genes. Cold stress induced SE-bound genes in both WT and *se* mutant plants, but the expression levels of most of these genes were lower under both non-stress and stress conditions in *se-1* compared to WT (*Figure 2—figure supplement 3*). These results indicate that SE is important for increasing the expression of these target genes, but not for their stress-dependent induction.

Because we found SE enhancing the expression of intronless genes, we asked whether the metazoan homologue of SE, ARS2, fulfills a similar function. For this, we analyzed the transcriptome of a *Drosophila melanogaster ars2* transposon insertion mutant (*Sabin et al., 2009*; *Garcia et al., 2016*). Interestingly, we found that genes down-regulated in *ars2* mutants contained significantly less introns when compared to all Drosophila genes, similarly to what we observed in Arabidopsis *se* mutants (*Figure 2J,K*). Interestingly, also the only suggested direct target of human ARS2, *SOX2*, turns out to be an intronless gene (*Andreu-Agullo et al., 2011*). These observations might suggest that SE's positive regulatory function on the expression of intronless genes might be conserved among eukaryotes.

## SERRATE promotes association of Ser5P and Ser2P pol II complexes to its targets genes

Because SE binds mainly to exonic regions of its target genes, we hypothesized that SE is involved in transcriptional regulation, likely by recruiting pol II or enhancing its association with a specific set of ILGs. Under this scenario, one would expect the association of pol II at SE target genes to be reduced in the *se* mutants. To investigate this notion, we conducted pol II ChIP-qPCR experiments and measured pol II levels at five different established SE-target genes. At all loci examined, pol II occupancy was lower in the *se* mutants than in WT, indicating that SE positively regulates the transcription of its target ILGs (*Figure 3A,B*, *Figure 3—figure supplement 1*). The reduced pol II occupancy at ILGs in the *se* mutants was not due to changes in pol II levels in the *se* mutants, as pol II protein levels were unchanged in these mutants compared to WT (*Figure 3—figure supplement 1*). Moreover, SE had no apparent influence on the stability of the RNAs generated by the SE-target genes, suggesting that SE primarily enhances the transcription of its targets (*Figure 3—figure supplement 2*).

In all eukaryotes, an unmodified C-terminal domain (CTD) of pol II has been associated with transcription initiation, whereas hyperphosphorylation of CTD's serine 5 (Ser5P) and Ser2P are characteristic of promoter clearance/pol II pausing and elongation, respectively. To determine the step of transcription at which SE acts, we performed pol II ChIP experiments using antibodies raised against Ser2- or Ser5-phosphorylated CTD and analyzed the distribution of pol II in the promoter region, around the TSS, and at the gene body in five different SE-target genes in the WT and *se* mutants. At all loci tested, all pol II isoforms associated to a lesser extent in the *se* mutants than in the WT (*Figure 3C,D*, *Figure 3—figure supplement 3*). These results suggest that SE promotes the association of paused and elongating pol II with the chromatin of its target loci.

## SERRATE interacts with Ser5- and Ser2-phosphorylated pol II complexes

The enhanced association of pol II isoforms at SE-bound ILGs might involve physical associations between the pol II complexes and SE. To test this, we performed immunoprecipitation experiments with antibodies raised against pol II CTD Ser5P- and Ser2P, and IgG (as a negative control) and identified interacting proteins by mass spectrometry. Several known pol II subunits were successfully

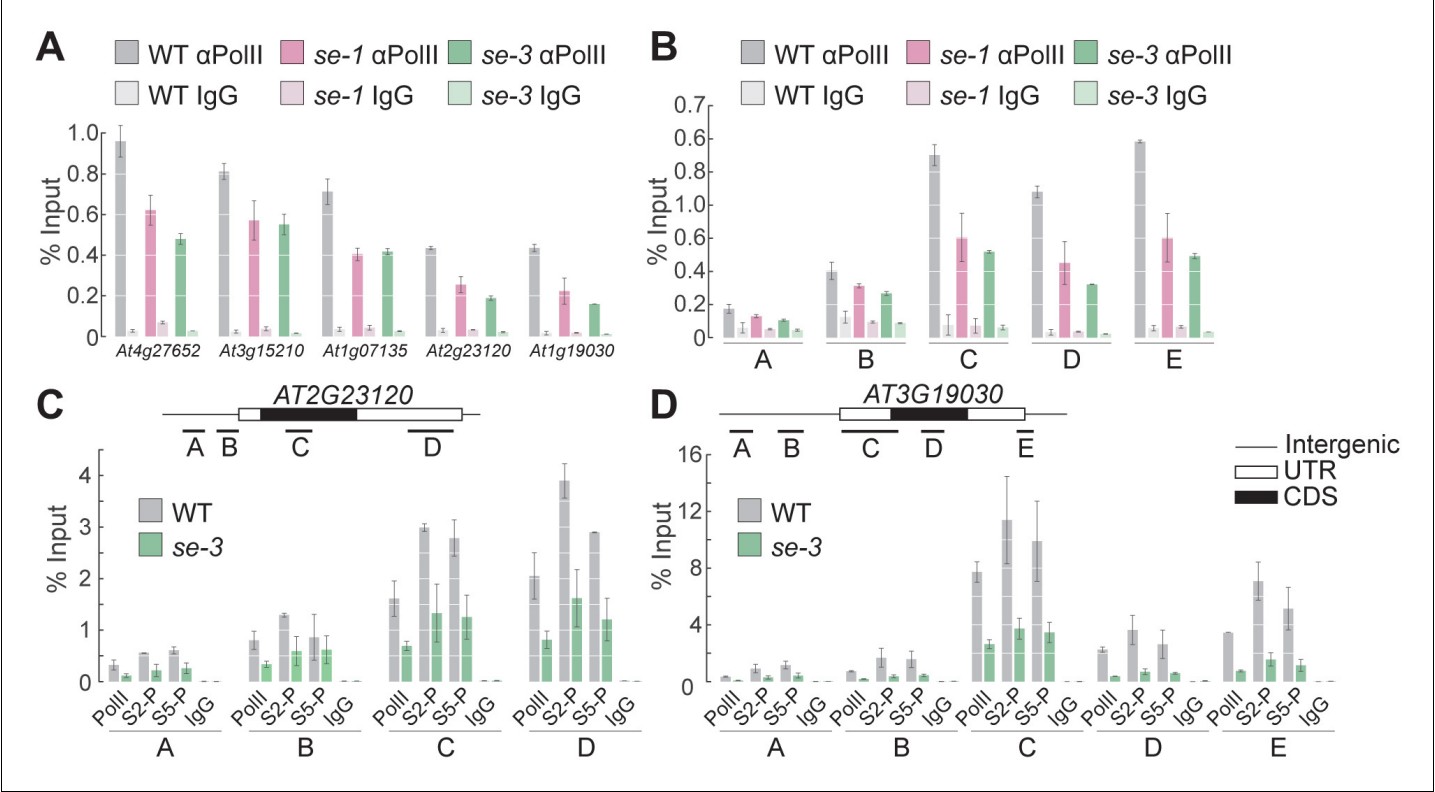

**Figure 3.** SE ensures efficient pol II association to intronless genes. (**A,B**) Analysis of pol II occupancy at SE target loci by pol II-ChIP qPCR in WT, *se-1* and *se-3* in gene bodies of five different genes (**A**) or at various genomic regions at *At3g19030* (**B**). Regions tested are depicted in panel D. A general pol II CTD antibody and mouse IgGs (as negative control) were used for immunoprecipitation. Additional gene loci were tested and are shown in *Figure 3—figure supplement 1*. Error bars indicate the range of two independent biological experiments. (**C,D**) Analysis of Ser5P and Ser2P pol II levels at SE target loci by ChIP qPCR in WT and *se-3*. General pol II CTD, pol II CTD Ser2P and pol II CTD Ser5P specific antibodies and mouse IgGs (as negative control) were used for immunoprecipitation. Additional gene loci were tested and are shown in *Figure 3—figure supplement 3*. Error bars indicate the range of two independent biological experiments.

DOI: https://doi.org/10.7554/eLife.37078.008

The following figure supplements are available for figure 3:

**Figure supplement 1.** SE is important pol II association to its target genes, but does not affect pol II levels.
DOI: https://doi.org/10.7554/eLife.37078.009

**Figure supplement 2.** The stability of RNAs produced by SE target genes is not affected SE.
DOI: https://doi.org/10.7554/eLife.37078.010

**Figure supplement 3.** SE ensures efficient association of pol II CTD-Ser2P and pol II CTD-Ser5P to its target genes.
DOI: https://doi.org/10.7554/eLife.37078.011

retrieved in all IP experiments (*Figure 4A*). In addition, peptides corresponding to SE were enriched in immunoprecipitations with antibodies against pol CTD Ser5P- and Ser2P (*Figure 4A*). These results reveal that SE is associated with Ser5- and Ser2-phosphorylated pol II complexes and it suggests that SE binds to pol II complexes during transcriptional pausing and elongation. We verified the interaction between SE and Ser5- and Ser2-phosphorylated pol II complexes by performing co-immunoprecipitation experiments using plant extracts from Arabidopsis and *Brassica oleracea* var. botrytis (*Figure 4B–E*). To visualize the localization and possible interactions between SE and pol II in plant nuclei, we performed immunolocalization analysis on 100 nm thin thawed cryosections with antibodies against the pol II CTD Ser5P, Ser2P and a triple HA-tagged version of SE driven by its own regulatory elements. Two independent super-resolution light microscopy techniques, Airyscan (*Figure 4F,G*) and super-resolution optical fluctuation (SOFI) imaging (*Figure 4H,I*) revealed that SE, pol II CTD Ser5P, and pol II CTD Ser2P each formed small, dotted speckles in the nucleus and that a subfraction of SE co-localized with the pol II CTD Ser2P and Ser5P speckles (*Figure 4F,G*). The finding that the majority of SE, pol II CTD Ser2P and Ser5P speckles did not co-localize is in line with our

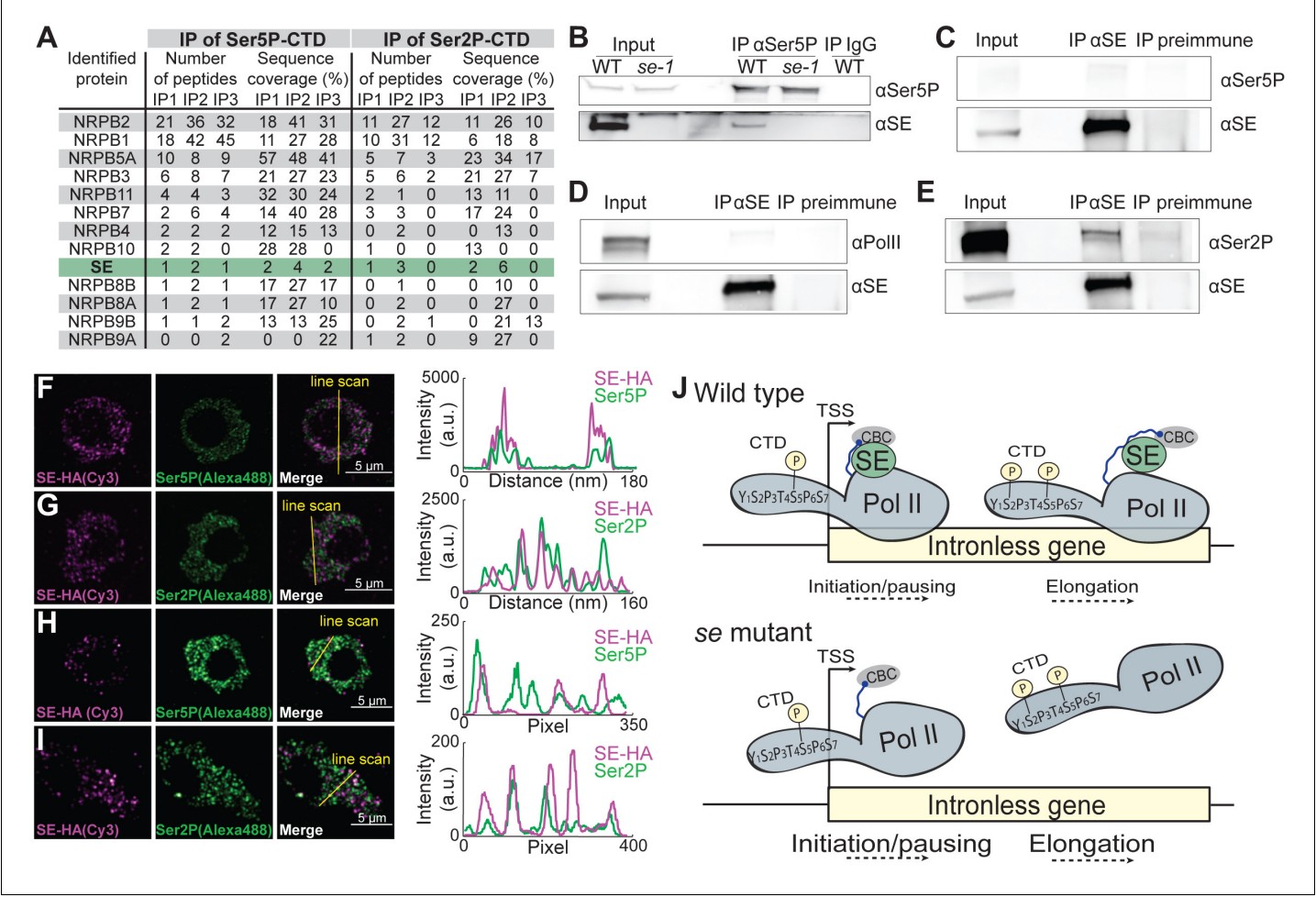

**Figure 4.** SE associates with Ser5P and Ser2P pol II complexes. (**A**) Summary of mass spectrometry (MS) analysis of immunoprecipitation (IP) reactions using antibodies against RNA polymerase II phosphorylated at Serine 2 (Ser2P) and Serine 5 (Ser5P) of its C-Terminal domain (CTD) from Arabidopsis lysates. Protein coverage and peptide number are represented for known RNA polymerase subunits and SE in three independent biological replicates. (**B–E**) Co-immunoprecipitation experiments using SE and phosphorylation-specific pol II antibodies. Western blot analysis of pol II Ser5P and IgG (as a negative control) immunoprecipitation experiments from Arabidopsis (**B**). Immunoprecipitation of SE from *B. oleracea* using a SE-specific antibody followed by detection of unphosphorylated (**D**), Ser5P (**C**) and Ser2P (**E**) pol II. The images are representative of at least three independent biological replicates. (**F–I**) Immunolocalization of pol II Ser5P, pol II Ser2P and SE-HA on a 100 nm thin thawed cryosection analyzed by super-resolution microscopy (F,G: Airyscan; H,I: Super-resolution optical fluctuation (SOFI) microscopy). Lines depicte the position for line blot analysis. (**J**) Proposed mechanism for the function of SE at intronless genes.

DOI: https://doi.org/10.7554/eLife.37078.012

The following figure supplements are available for figure 4:

**Figure supplement 1.** Analysis of SE and AGO1 targets.
DOI: https://doi.org/10.7554/eLife.37078.013

**Figure supplement 2.** Analysis of SE target genes down-regulated in *se* mutants.
DOI: https://doi.org/10.7554/eLife.37078.014

observation that SE associates only with a small subfraction of transcribed genes (*Figure 1*). Together, these results indicate that SE binds to Ser5- and Ser2-phosphorylated pol II complexes, and that SE associates with these complexes during pausing and elongation of pol II to enhance the expression of a subset of Arabidopsis ILGs (*Figure 4J*).

## Summary and perspectives

The main findings presented in this study support a novel mechanism that explains how a subset of Arabidopsis genes without introns is expressed at high levels. This subset of genes represents a clear exception to the trend conserved across kingdoms, wherein low intron number in a gene is associated with a low expression level (*Le Hir et al., 2003*; *Narsai et al., 2007*; *Jeffares et al., 2008*). At these Arabidopsis loci, SE is recruited to the nascent RNAs through interaction with the CBC and specifically enhances the association with Ser5P- and Ser2P pol II complexes (*Figure 4J*). Interestingly, another Arabidopsis miRNA player, AGO1, was recently shown to associate with chromatin and is guided to specific genes via small RNAs (*Liu et al., 2018*). SE bound to 8.5% of the described AGO1 target genes (*Figure 4—figure supplement 1*). These observations reveal that the majority of SE target genes are not bound by AGO1, but it also raises the exciting possibility that SE and AGO1 simultaneously bind and regulate a small set of target genes.

Although RNA production and the CBC play an important role for SE binding to its target genes, recruitment of SE to specific genes might also involve other trans-regulatory factors, such as specific transcription factors, that recognize motifs enriched in the proximity of SE binding sites (*Supplementary file 2*). Interestingly, a known SE interactor, C-TERMINAL DOMAIN PHOSPHA-TASE-LIKE1 (CPL1), interacts with transcription factors, suggesting that specific DNA-binding proteins might also help recruit SE or SE-interacting proteins to specific genomic loci (*Manavella et al., 2012*; *Guan et al., 2014*). Notably, CPL1 acts as a transcriptional repressor (*Xiong et al., 2002*), suggesting that SE might also interact with repressors of transcription at specific loci. Indeed, some SE-targets are upregulated in the *se* mutant (*Figure 4—figure supplement 2*), suggesting that SE can also execute repressive functions at some gene loci. Further characterization of the binding sites of SE and its metazoan ortholog ARS2 and their effects on gene expression will provide important insights into the coupling of transcription and RNA processing in eukaryotes.

Our findings have implications for biotechnology, as SE-dependent activation of gene expression might be utilized to increase the expression of specific transgenes and endogenous genes. Whether similar mechanisms or mechanisms acting in parallel to regulate the expression of other ILGs remains to elucidated. The fact that ILGs are present in all eukaryotic genomes and often fulfill important functions (*Grzybowska, 2012*) suggests that additional regulatory mechanisms exist to ensure the coordination of gene expression in the absence of introns and splicing.

## Materials and methods

**Key resources table**

| Reagent type (species) or resource | Designation | Source or reference | Identifiers | Additional information |
|---|---|---|---|---|
| Strain, strain background (A. thaliana) | se-1 mutant | Clarke et al. (1999) | NASC_ID:N3257 | |
| Strain, strain background (A. thaliana) | se-3 mutant | Grigg et al. (2005) | SALK_083196 | |
| Strain, strain background (A. thaliana) | cbp20-1 mutant | Papp et al. (2004) | | |
| Strain, strain background (A. thaliana) | SE_Pro:SE-3HA | This study | | |
| Antibody | αSE | This study | | Raised against the peptide: QDLDAPEE EVTVIDYRSL, 10 µl/ChIP; 1:300 (IP); 1:2000 (WB) |
| Antibody | αCDT | Abcam | ab817 | 10 µl/ChIP; 1:250 (IP); 1:1000 (WB) |
| Antibody | αCTDI-S2P | Abcam | ab5095 | 10 µl/ChIP; 1:250 (IP); 1:1000 (WB) |
| Antibody | αCTDI-S2P | Abcam | ab5131 | 1:200 (IF) |
| Antibody | αCTDI-S5P | Abcam | ab5408 | 10 µl/ChIP; 1:250 (IP); 1:2000 (WB); 1:200 (IF) |
| Antibody | rat monoclonal IgG_1 3F10 anti-HA | Sigma | 000000011867423001 | 1:25 (IF) |

*Continued on next page*

*Continued*

| Reagent type (species) or resource | Designation | Source or reference | Identifiers | Additional information |
|---|---|---|---|---|
| Antibody | goat anti-rat IgG coupled to Cy3 | Dianova | 112-165-167 | 1:400 (IF) |
| Antibody | goat anti-rabbit antibody coupled to Alexa488 | Dianova | 111-546-047 | 1:400 (IF) |
| Commercial assay or kit | MinElute Reaction Cleanup Kit | Qiagen | 28204 | |
| Commercial assay or kit | QuantiNova SYBR Green PCR Kit | Qiagen | 208052 | |
| Commercial assay or kit | SYBR-Green Maxima | Thermo Fisher Scientific | K0251 | |
| Commercial assay or kit | RevertAid First Strand cDNA Synthesis Kit | Thermo Fisher Scientific | K1621 | |
| Commercial assay or kit | RNeasy Plant Mini Kit | Qiagen | 74904 | |
| Commercial assay or kit | TruSeq ChIP Library Preparation Kit | Illumina | IP-202–1012 | |
| Commercial assay or kit | NEBnext poly(A) mRNA Magnetic Isolation Module | New England Biolab | #7490 | |
| Commercial assay or kit | ScriptSeq v2 RNA-Seq Library Preparation Kit | Illumina, | SSV21124 | |
| Chemical compound, drug | mouse IgG | Santa Cruz | sc-2025 | |
| Chemical compound, drug | rabbit IgG | Santa Cruz, | sc-2027 | |
| Chemical compound, drug | Agarose A/G-Plus beads | Santa Cruz | sc-2003 | |
| Chemical compound, drug | protease inhibitor cocktail | SIGMA | P9599 | |
| Chemical compound, drug | Phosphatase Inhibitor Cocktail 2 | SIGMA | P5726 | |
| Chemical compound, drug | Phosphatase Inhibitor Cocktail 23 | SIGMA | P0044 | |
| Chemical compound, drug | protein A-Agarose | Roche | 11134515001 | |
| Chemical compound, drug | protein G-Agarose | Roche | 11243233001 | |
| Chemical compound, drug | cOmplete, EDTA-free | Sigma | 000000011873580001 | |

## Plant material and growth conditions

The Arabidopsis mutant plants, *se-1*, se-3 and *cbp20-1*, used in this study have been described before (*Clarke et al., 1999*; *Papp et al., 2004*; *Grigg et al., 2005*). All mutant plants are in the Columbia (Col) background which served as a wild-type control in all experiments. For the expression of a 3xHA-tagged SE in *se-1* mutant backgrounds, we PCR-amplified the *SE* genomic locus (2244 bp upstream of the ATG start codon till the last coding triplet before the stop codon) and cloned it into Topo GW8 (Life Technologies). The *SE* genomic clone was transferred into pGWB413 to generate a *SE$_{Pro}$:SE-3HA* construct, which was introduced into *se-1* mutants by floral dip transformation. The majority of the obtained T1 transgenic plant rescued the developmental phenotypes of *se-1* mutants and independent single insertion lines were propagated for further downstream analyses. Plants were grown for 10 and 16 days on solid or 6 days in liquid ½ strength Mourashige and Skoog (Duchefa) media at 22°C under constant light conditions.

## Stress treatments

Cold stress was applied by transferring 10-day-old seedlings grown on the plates to a container with (stress treatment) or without (control treatment) ice and incubated for the indicated time in the growth chamber before harvesting the samples.

## RNA stability assay

The RNA stability assay was performed on 6-day-old seedlings grown in liquid culture. The seedlings were transferred to liquid ½ strength Mourashige and Skoog media with or without 200 µg/ml cordycepin (Sigma) and incubated for the indicated time before harvesting the samples. RNA extraction, cDNA synthesis and steady-state transcript level were analyzed as described in section 'RNA isolation and analysis'.

## RNA isolation and analysis

Total RNA was extracted using the RNeasy Plant Mini Kit (Qiagen) according to manufacturer's instructions. For quantitative RT-PCR analyses, 1 µg of total RNA was treated with DNAse and reverse transcribed using RevertAid First Strand cDNA Synthesis Kit (Thermo Fisher Scientific) according to the manufacturer's instructions. Oligo-dT primers were added for reverse transcription of mRNAs. Quantitative PCR was performed in reactions containing SYBR-Green (Maxima, Thermo Fisher Scientific or QuantiNova, Qiagen) on a CFX384 system (Bio-Rad). All measurements were repeated twice or three times, all experiments were performed using at least three biological replicates. All qPCR runs were performed in the presence of a standard curve of amplification. *PP2A* served as a normalization control for all experiments. Relative expression was calculated using the ∆∆ct method ($2^{-\Delta\Delta ct}$). All oligonucleotides are listed in *Supplementary file 3*.

## Chromatin-immunoprecipitation (ChIP)

16-day-old seedlings were harvested on ice and crosslinked with 1% (v/v) formaldehyde in MQ-buffer (10 mM Sodium-Phosphat-buffer pH 7, 50 mM NaCl, 100 mM sucrose, 1% (v/v) fromaldehyde) for 20 min using vacuum infiltration. Subsequently, the reaction was quenched by adding glycine to a final concentration of 125 mM. Nuclei enrichment was performed on 1.5 g or 3 g of cross-linked plant material using the HONDA buffer. For the ChIP experiments in presence of cordycepin, seedlings were grown in half strength Mourashige and Skoog liquid medium on a shaker (120 rpm) in constant light for six days and were treated for 2 hr with 200 µg/ml of cordycepin. For the treatment 6 flasks of 25 mg seeds/50 ml liquid media were pooled per sample and processed as described above.

Ground plant material was resuspended in 30 ml HONDA buffer (0.4 M sucrose, 1.25% (w/v) Ficoll, 2.5% (w/v) Dextran T40, 25 mM HEPES, pH 7.4, 10 mM $MgCl_2$, 0.5% (v/v) Triton X-100, 1 mM phenylmethylsulfonyl fluoride (PMSF), 10 mM dithiothreitol (DTT), cOmplete, EDTA-free (Roche)). Homogenate was filtered through two layers of Miracloth. The Miracloth was washed using 10 ml of HONDA buffer. Nuclei were precipitated by centrifugation (1500 g, 15 min, 4°C). The pellet was washed five times using 1 ml of HONDA buffer. After each washing step nuclei were collected by centrifugation (1000 g, 5 min, 4°C). The nuclei pellet was washed in 1 ml of M3 buffer (10 mM sodium phosphate buffer, pH 7, 100 mM NaCl, 10 mM DTT, cOmplete, EDTA-free (Roche)) and nuclei were collected by centrifugation (1000 g, 5 min, 4°C).

The enriched nuclei pellet was resuspended in 1 ml sonic buffer (10 mM sodium phosphate buffer, 100 mM NaCl, 1% sarkosyl, 10 mM ethylenediamine tetraacetic acid (EDTA), 1 mM 4-(2-aminoethyl)benzenesulfonyl fluoride PEFAbloc, complete proteinase inhibitors (Roche)) and sonicated to a fragment size of 600–250 bp using the sonicator Covaris E220 (duty cycle: 20%, peak intensity: 140, cycles of burst: 200, time: 2.5 min). After centrifugation (14000 g, 5 min, 4°C), 700 µl supernatant was mixed with 1 Vol of IP buffer (10 mM HEPES, pH 7.4, 150 mM KCl, 5 mM $MgCl_2$, 10 µM $ZnSO_4$, 1% (v/v) Triton X-100, 0.05% (w/v) sodium dodecyl sulfate (SDS)). 140 µl (20% of the IP) was stored as input control. Antibodies were added to the IP fractions (10 µl specific SE antibody (raised against the 18AA C-terminal peptide, Agrisera, Sweden)), 10 µl pol II (ab817, abcam), 25 µl mouse IgG (Santa Cruz), 10 µl pol II-S2P (ab5095, abcam), 10 µl pol ll-S5P (ab5408, abcam), 25 µl normal IgG rabbit (Santa Cruz)) and incubated over night at 4°C on a rotating wheel. 40 µl of pre-blocked Agarose A/G-Plus beads (Santa Cruz, sc-2003) were added to the IP fraction and incubated for 6 hr at 4°C. The beads were washed five times using IP buffer for 10 min at 4°C followed by a centrifugation step (14000 g, 2 min, 4°C). Precipitated protein-DNA complexes were eluted form the beads by three times elution with 100 µl acidic glycine elution buffer (100 mM glycine, 500 mM NaCl, 0.005% (v/v) Tween-20, pH 2,8). Each elution fraction was immediately neutralized by transferring it to a tube containing 150 µl 1M TRIS, pH 9. All next steps were performed with both input and IP samples. The samples were treated with 1 µl of RNase A (Thermo Scientific) for 15 min at 37°C followed by a treatment with 1.5 µl Proteinase K (Roche) overnight at 37°C. The material was reverse cross-linked by an additional treatment of 1.5 µl Proteinase K and an incubation step at 65°C for 6 hr. DNA was purified using the MinElute Kit (Qiagen) according to manufacturer's instructions. The precipitated DNA was used to prepare DNA sequencing libraries or to quantify enriched DNA fragments by standard qPCR methods. Enrichment of DNA fragments for SE and pol II ChIP analysis were calculated as % input ($2^{(ct\ input\ adjusted\ -\ ct\ IP)} \times 100$).

## ChIP-seq Library preparation, sequencing and data analysis

Three independent ChIP experiments performed on 16-day-old seedlings from Col-0 (target samples) and from *se-1* mutants (control samples) were used to prepare ChIPseq libraries using the TruSeq ChIP Library Preparation Kit (Illumina) according to manufacturer's instructions. Purification of ligation products was achieved on a BluePippin (Sage Sciences) using a 1.5% Agarose cassette. Elution size was 200–500 bp. All libraries were sequenced on HiSeq3000 using 150 bp single ends kit. The output was between 56 and 70 million per library. Sequencing data have been deposited with the European Nucleotide Archive (ENA), accession number ERP016859. Trimmed reads were aligned against Arabidopsis genome (TAIR10 release) using bwa. The output was used to generate BAM files required for calling peaks. Unique mapped reads were used to call peaks using MACS2. Peaks were called for the three biological replicates independently: sequences from *se-1* samples from each replicate was used as control for each Col-0. Overlapping peaks were called using the R package DiffBind (http://bioconductor.org/packages/release/bioc/html/DiffBind.html) and it results in 1012 peaks represented in all biological replicates. These overlapping peaks were considered highly reliable and used for further analysis.

Nucleotide sequences for the 1012 peaks subset were extracted and used to peaks annotation analysis using ChIPseek (*Chen et al., 2014*). The same subset of sequences was analyzed for conservative binding site RSAT (*Supplementary file 2*) (*Medina-Rivera et al., 2015*).

## RNA-seq Library preparation, sequencing and data analysis

For library preparation, RNA was isolated from WT and *se-1* plants harvested and treated as for the ChIP-seq analysis (without FA crosslinking, see above). Total RNA was isolated using the RNeasy Plant Mini Kit (Qiagen) according to the manufacturer's instructions. mRNA was isolated using the NEBnext poly(A) mRNA Magnetic Isolation Module (New England Biolabs, #7490), followed by library preparation using ScriptSeq v2 RNA-Seq Library Preparation Kit (Illumina). For library preparation mRNA was used as input starting at the fragmentation step in the protocol (step number 5).

The mRNA libraries were sequenced on an Illumina HiSeq3000 using paired-end sequencing of 150 bp in length. Sequence quality was assessed using FASTQC (v0.10.1) and rRNA and tRNA sequences were filtered out and aligned with Bowtie2 (*Langmead and Salzberg, 2012*) version 2.1.0. The Bowtie2 –un-conc-gz parameter was adjusted to gain unmapped reads. Filtered reads were aligned to the genome (*Kersey et al., 2016*) (*Arabidopsis thaliana* Ensembl Plants release 34 bin, ftp://ftp.ensemblgenomes.org/pub/release-34/plants/fasta/arabidopsis_thaliana/dna/Arabidopsis_thaliana.TAIR10.dna.toplevel.fa.gz, ftp://ftp.ensemblgenomes.org/pub/release-34/plants/gtf/arabidopsis_thaliana/Arabidopsis_thaliana.TAIR10.34.gtf.gz) with TopHat2 (*Kim et al., 2013*), version 2.1.1 using the following parameters: tophat2 -p 10 -i 10 -I 1000 –library-type fr-secondstrand -G (alignment was guided with the GTF annotation file) and after mapping raw read counts were collected with featureCounts. DEseq2 (*Love et al., 2014*) was used to identify the genes affected *se-1* mutation. The genes with a padj < 0.05 were considered to be repressed and induced in se-1 mutants compared to WT. Raw data were deposited in the Gene Expression Omnibus (GEO) under accession number GSE99367.

## Analysis of SE target genes

Data sets for gene expression values were obtained from *Laubinger et al. (2008)*, and *Zeller et al. (2009)*, epigenetic data sets were obtained from *Liu et al. (2016)* (*Zeller et al., 2009*; *Liu et al., 2016*; *Bi et al., 2017*). Intron numbers were calculated based on TAIR10, isoform. 1, because in most cases isoform. one is the most abundant isoform. AGO1 ChIP-seq data were described in *Liu et al. (2018)*. Intron numbers, gene expression values and epigenetic data were compared for SE-target and non-targets and a Wilcoxon-Mann-Whitney test was applied for statistical analyses.

## Analysis of intron number in plant stress responses

A pipeline was developed to generate information about intron number in genes that are induced by stress in different plant species, including Arabidopsis, rice, soy and maize. In brief, the 'intron number' in each gene was computed with an in-house script based on Ensembl genome annotation files (*Kersey et al., 2016*) and a list of genes induced by stress was generated using expression information from publically available RNA-seq data. The list of stress-induced genes was merged

with the intron number list. The median of the 'intron number' distribution among the stress-induced gene list was compared with the median of the 'intron number' distribution of all genes in each corresponding genome employing a two sample Wilcoxon test.

Arabidopsis stress data sets were obtained from *Zeller et al. (2009)*. The rice RNA-seq dataset with SRA accession number DRA000959 (DDBJ Center) was used to generate a list of stress-induced genes in rice (*Kawahara et al., 2016*). The rice reference genome used for the alignment and intron number calculations was downloaded from Emsembl, release 32 bin (ftp://ftp.ensemblgenomes.org/pub/plants/release-32/fasta/oryza_sativa/dna/Oryza_sativa.IRGSP-1.0.dna.toplevel.fa.gz, ftp://ftp.ensemblgenomes.org/pub/plants/release-32/gtf/oryza_sativa/Oryza_sativa.IRGSP-1.0.32.gtf.gz).

Maize stress RNA-seq dataset were obtained from NCBI GEO accession GSE76939 (https://www.ncbi.nlm.nih.gov/geo/query/acc.cgi?acc=GSE76939) (*Lu et al., 2017*). Maize annotations were downloaded from the official Ensembl website, release 32 bin (ftp://ftp.ensemblgenomes.org/pub/plants/release-32/fasta/zea_mays/dna/Zea_mays.AGPv4.dna.toplevel.fa.gz ftp://ftp.ensemblgenomes.org/pub/plants/release-32/gtf/zea_mays/Zea_mays.AGPv4.32.gtf.gz).

Mapping and normalization and list of differentially expressed genes for rice and maize RNA-seq datasets were carried out as described above. A list of stress-induced genes was generated where log2 [stress/mock]>=2 and padj < 0.05) was filtered out.

The dataset expression tables S2, S3, S5, S7 from a published study (*Belamkar et al., 2014*) were directly used to generate a list of stress-induced genes in by filtering out upregulated genes (log2 [stress/mock]>=2 and padj < 0.05). The reference genome used to calculate 'intron number' in soy was ftp://ftp.ensemblgenomes.org/pub/plants/release-32/gtf/glycine_max/Glycine_max.V1.0.32.gtf.gz.

## Analysis of Drosophila ars2 mutant transcriptomes

For the analysis of the Drosophila ars2 mutant transcriptome, we obtained published data sets described in *Garcia et al. (2016)*. Adapters and low quality bases were removed using TRIMMOMATIC, version 0.36 (*Bolger et al., 2014*), with the following parameters ILLUMINACLIP:TruSeq3-PE-2.fa:2:30:10 LEADING:3 TRAILING:3 SLIDINGWINDOW:4:15 MINLEN:35. rRNA and tRNA sequences were filtered out from the Ensembl 91 and extracted using the tophat2 gtf_to_fasta (*Zerbino et al., 2018*). Addtitonal tRNA sequences were for filtering were retrieved from GtRNAdb (http://gtrnadb.ucsc.edu/GtRNAdb2/genomes/eukaryota/Dmela6/dm6-tRNAs.fa) (*Chan and Lowe, 2016*). Filtered reads were aligned against the *Drosophila melanogaster* genome, using the corresponding GTF annotation file (Ensembl 91 release) (*Zerbino et al., 2018*) with default parameters using Tophat2, version 2.1.1 (*Kim et al., 2013*). Read counts were collected with featureCounts from the Subread package version v1.5.2 (*Liao et al., 2013*) using featureCounts –T 6 R -p -F GTF -J -M -G -a. To collect expression values, DESeq2 (version 1.16.1) (*Love et al., 2014*) was executed according to the authors' recommendations, but with using the parameter 'contrast' with the wild-type sample. Downregulated genes (padj < 0.05 and fold change >= 2) were filtered out and merged together with the corresponding isoforms' intron number information based on the GTF file. Final intron number distributions from all annotated genes and downregulated genes in the ars2 mutant were plotted using ggplot2, version 2.1.0 (*Wickham, 2009*), and a two sample Wilcoxon test was performed with ggsignif (version 0.4.0) (*Ahlmann-Eltze, 2017*). Similarly, intron numbers of genes downregulated (padj < 0.05 and fold change >= 2) in the Arabidopsis se-1 mutant were compared to all annotated genes (TAIR10, omitting transposons and pseudogenes).

## Sample preparation for mass spectometry

For RNA polymerase II immunoprecipitation, WT Arabidopsis seedlings were harvested and kept on ice during harvesting/weighing time following by flash freezing in liquid nitrogen. 20 g of frozen material were ground together with liquid nitrogen using a pre-cooled mortar and pestle, and liquid nitrogen was added five times until a fine powder was obtained. 20 mL of extraction buffer (50 mM HEPES buffer pH = 7.4, 50 mM NaCl, 1 mM phenylmethylsulfonyl fluoride (PMSF), 1 mM dithiothreitol, 10% (v/v) glycerol, 1% (v/v) IGEPAL CA-630, 1:100 protease inhibitor cocktail (SIGMA, P9599), 1:500 Phosphatase Inhibitor Cocktail 2 (SIGMA, P5726), 1:500 Phosphatase Inhibitor Cocktail 3 (SIGMA, P0044) and cOmplete, EDTA-free (Roche)) was added and mixed with the plant samples. The resulting lysate was transferred to reaction tubes and spun at 13.000 rpm for 15 min at 4°C. The

supernatant was passed through a 0.22 µM filter. 5 mL of lysate were used per IP and incubation was carried out in 5 mL Eppendorf tube – low protein binding. 20 µg of each antibody were pre-bound to 50 µL protein A-Agarose (for antibodies raised in rabbit) or protein G-Agarose (for antibodies raised in mouse) beads (Roche, 11134515001 and 11243233001, respectively) in extraction buffer, at 4°C for 2 hr on a rotating wheel. An antibody specifically recognizing the C'-terminal domain (CTD) of RNA polymerase II (pol II) (abcam, ab817) was used for immunoprecipitation. Antibodies recognizing pol II CTD Ser2P (abcam, ab5095) and pol II CTD Ser5P (abcam, ab5408) were used for immunoprecipitation of phosphorylated CTDs. 20 µg of mouse IgG or rabbit IgG antibodies were used as negative controls. Incubation with beads was carried out at 4°C for 30 min on a rotating wheel. After incubation, beads were spun at 1000 g at 4°C for 2 min, supernatant was removed and a 5 min wash with 1 mL of extraction buffer was carried out at 4°C on a rotating wheel. This washing step was repeated five times. Proteins were eluted from the beads in 30–35 µL of 2X Laemmli Buffer at 80°C for 10 min and stored overnight at 4°C. Eluted protein samples were purified using SDS PAGE (Invitrogen). Coomassie-stained gel pieces were excised and in-gel digested using Trypsin as described previously (*Borchert et al., 2010*). Extracted peptides were desalted using C18 StageTips and subjected to LC-MS/MS analysis (*Rappsilber et al., 2007*). LC-MS/MS analyses were performed on an Easy nano-LC (Thermo Scientific) coupled to an LTQ Orbitrap XL mass spectrometer (Thermo Scientific) as decribed elsewhere (*Franz-Wachtel et al., 2012*). The peptide mixtures were injected onto the column in HPLC solvent A (0.1% formic acid) at a flow rate of 500 nl/min and subsequently eluted with an 127 min segmented gradient of 5–33-50–90% of HPLC solvent B (80% acetonitrile in 0.1% formic acid) at a flow rate of 200 nl/min. The 10 most intense precursor ions were sequentially fragmented in each scan cycle using collision-induced dissociation (CID). In all measurements, sequenced precursor masses were excluded from further selection for 90 s. The target values were 5000 charges for MS/MS fragmentation and $10^6$ charges for the MS scan.

## Mass spectrometry data processing

The MS data were processed with MaxQuant software suite v.1.5.2.8 (*Cox and Mann, 2008*). Database search was performed using the Andromeda search engine, which is integrated in MaxQuant (*Cox et al., 2011*). MS/MS spectra were searched against a target-decoy Uniprot database consisting of 33,431 protein entries from *A. thaliana* and 285 commonly observed contaminants. In database search, full specificity was required for trypsin. Up to two missed cleavages were allowed. Carbamidomethylation of cysteine was set as fixed modification, whereas oxidation of methionine, acetylation of protein N-terminus and phosphorylation of serine, threonine and tyrosine were set as variable modifications. Initial mass tolerance was set to 4.5 parts per million (ppm) for precursor ions and 0.5 dalton (Da) for fragment ions. Peptide, protein and modification site identifications were reported at a false discovery rate (FDR) of 0.01, estimated by the target/decoy approach (*Elias and Gygi, 2007*). The iBAQ algorithm was enabled to estimate quantitative values by dividing the sum of peptide intensities of all detected peptides by the number of theoretically observable peptides of the matched protein (*Schwanhäusser et al., 2011*). The mass spectrometry proteomics data have been deposited to the ProteomeXchange Consortium via the PRIDE partner repository with the dataset identifier PXD006004 (*Vizcaíno et al., 2014*). (*Reviewer account details:* **Username:** reviewer88206@ebi.ac.uk **Password:** fcjpzeqK). All proteins identified in our experiments are listed in *Supplementary file 4*.

## Sample preparation for co-immunoprecipitation

For Ser5P RNA polymerase II, immunoprecipitation protocol was identical to the mass spectroscopy sample preparation, except that 3 g of Arabidopsis of WT or *se-1* seedlings per sample were processed individually with 4 mL of extraction buffer and 40 µL of Protein G-Agarose beads (Roche, 11243233001) were used. For SE immunoprecipitation from cauliflower, 140 g of commercially available cauliflower was flash frozen in liquid nitrogen. Frozen samples were blended with a Rommelsbacher Floormixer MXH 1500, 1500 Watt blender together with 200 mL extraction buffer (20 mM Tris-HCl buffer pH 7.4, 150 mM NaCl, 1 mM PMSF, 1 mM ethylenediamine tetraacetic acid (EDTA), 5% (v/v) glycerol). Lysate was filtered through 2 layers of Miracloth (475855 - EMD Millipore). 1% (v/v) IGEPAL CA-630 (SIGMA), 1:100 protease inhibitor cocktail for plant cell lysate (SIGMA, P9599), 1:500 Phosphatase Inhibitor Cocktail 2 (SIGMA, P5726), 1:500 Phosphatase Inhibitor Cocktail 3

(SIGMA, P0044) and 1x cOmplete, EDTA-free (Roche) was added to samples. The lysate was incubated at 4°C for 30 min in a rotating wheel. 1.5 mL of lysate per sample were incubated with 5 µL of anti-SE antibody or 5 µL pre-immune serum from the same rabbit for 30 min at 4°C for on a rotating wheel. 20 µl of Agarose A/G-Plus beads (Santa Cruz, sc-2003) were added and samples were incubated for 1 hr. After incubation, beads were spun at 1000 g at 4°C for 2 min. Supernatants were removed and beads were washed five times using 1 mL of washing buffer (20 mM Tris-HCl buffer pH 7.4, 150 mM NaCl, 1 mM EDTA, 5% (v/v) glycerol, 1% (v/v) IGEPAL CA-630 (SIGMA)) at 4°C. Samples were eluted in 30–35 µL of 2X Laemmli buffer at 80°C for 10 min.

For both Arabidopsis and Cauliflower, samples were stored at −20°C until immunoblotting analysis using SE, Ser2P (ab5095), Ser5P (ab5408) and CTD (ab817) –specific antibodies.

## Sample preparation for TEM and super-resolution LM

Root tips of 4 day old Arabidopsis seedlings were fixed with 4% formaldehyde in MTSB (microtubule stabilizing buffer: 50 mM Pipes, 5 mM EGTA and 5 mM $MgSO_4$, pH 7) for 30 min, followed by fixation with 8% formaldehyde for another 90 min at ambient temperature. Thereafter, root tips were embedded in 10% gelatin and infiltrated with a mixture of sucrose and polyvinylpyrrolidone (PVP) (1.8 M sucrose (Merck)/20% PVP (Sigma, PVP10-100G). Infiltrated root tip pieces were mounted on metal stubs and frozen in liquid nitrogen. Ultrathin (100 nm) cryosections were cut at −115°C using a Leica UC7/FC7 cryoultramicrotome (Leica). Thawed sections were transferred to coverslips for immunofluorescence labelling or to pioloform and carbon coated electron microscopic grids for immunogold labelling.

*Immunolabelling:* Unspecific binding sites of thawed cryosections were blocked with 0.5% bovine serum albumin, 0.5% milk powder in phosphate-buffered saline (PBS) for 20 min. Antibody incubation was performed for 60 min (immunogen affinity purified rabbit antibody against pol II Ser5P ab5095 (1:200; Abcam), immunogen affinity purified rabbit antibody against pol II Ser2P ab5131 (1:200; Abcam), rat monoclonal IgG$_1$ 3F10 anti-HA (1:25; Roche/Sigma-Aldrich), diluted in blocking buffer). For double-labelling experiments, a mixture of the respective antibodies was used. After washing in blocking buffer for 20–30 min, sections were incubated with fluorescence (goat anti-rat IgG coupled to Cy3 (1:400), goat anti-rabbit antibody coupled to Alexa488 (1:400), (all markers were from Dianova, Germany), all markers diluted in blocking buffer) for 60 min. After final washing for 20–30 min in blocking buffer and PBS, sections mounted on coverslips were stained for DNA (4',6-diamidino-2-phenylindole, DAPI))(1 mg/ml; Sigma) and embedded in Mowiol 4.88 (Calbiochem) containing the anti-fading reagent DABCO (25 mg/ml; Sigma), whereas sections mounted on grids were silver-enhanced for 30 min using R-Gent (Aurion), washed thoroughly with double distilled water, stained with 1% uranyl acetate and finally embedded in a thin layer of 1.8% methyl cellulose (catalogue no. M-6385; Sigma-Aldrich) containing 0.3–0.45% uranyl acetate. The following immunofluorescence controls were performed: (i) anti-HA labelling on WT root tips resulted in low background, (ii) labeling with other pol II-specific antibodies show a similar labeling pattern, however with higher unspecific background, (iii) omitting first antibodies resulted in negligible labeling.

## Confocal laser scanning microscopy (Airyscan)

A Zeiss LSM 880 equipped with Airyscan detector was used. In order to increase axial resolution, immunolabelled thawed cryosections of 100 nm thickness were analyzed. Sections were imaged with a C-Plan-Apochromat 63x/1.40 Oil DIC UV-VIS-IR M27 objective. For the Airyscan mode, optimal frame size settings were used (pixel size is 40 nm). Images were processed using Airyscan processing default settings. Laser wavelength was 488 nm (~1%) and 561 nm (~1%), binning mode 1 × 1, detector gain 800, beam splitter MBS 488/561, Cy3 filter BP 495–550 + LP 570, Alexa488 filter BP 420–480 + BP 495–550, averaging factor 2. Contrast and brightness was adapted.

## Super-resolution optical fluctuation imaging (SOFI)

We also employed Super-resolution Optical Fluctuation Imaging (SOFI) as a microscopy technique to enhance the contrast of neighboring or overlapping protein clusters, which proved to be insufficient in classical light microscopy. This method relies on higher-order statistical analysis of temporal fluctuations (caused by fluorescence blinking/intermittency) recorded in a sequence of images (movie) (*Dertinger et al., 2009*). While the absolute resolution enhancement largely depends on the

sample and fluorophores, it is especially efficient in suppressing background signals which further improves image contrast. Also for this type of analysis, immunolabelled thawed cryosections of 100 nm thickness were used to obtain a high z resolution.

To enhance and control the blinking behavior of the organic dye labels Cy3 and ATTO488, we used a common super-resolution imaging buffer consisting of an oxygen scavenger system (glucose, glucose oxidase, catalase) and beta-mercaptoethanol in a Tris/NaCl buffer of pH 8.0.

## Microscope setup, acquisition and data analysis for SOFI

Movies were taken on a custom-built super-resolution microscope with total internal reflection (TIRF) illumination, using an Orca-Flash4.0 Digital CMOS camera (Hamamatsu, Japan). Fluorescence excitation of structures tagged with Cy3 were performed by a 561 nm laser (50 mW, Vortran Stradus, Laser2000, Germany), which was first spectrally cleaned up by a band pass filter (BrightLine 561/14, Semrock, USA), then spatially cleaned up by a glass fiber (PM-S405-XP, Thorlabs, USA) before focusing the beam onto the back focal plane of a high NA objective (Alpha Plan-Fluar 100x/1.49, Zeiss, Germany) with an off-axis lens to achieve TIRF illumination. Analogously, ATTO488-tagged structures were excited with a 488 nm laser (100 mW, Oxxius LBX, Laser2000, Germany), cleaned up by a band pass filter (ZET405/488x, Chroma, USA). Additionally, a 405 nm laser (80 mW, NANO 250, Qioptiq, UK) was used to further control the blinking behavior of the fluorescent tags in both cases. A quad-line beam splitter (zt450/488/561/640rpc, Chroma, USA) between the TIRF lens and the objective was used to reflected all illumination laser lines onto the sample plane while the respective fluorescence emission was able to pass through to the detection path. Here, further emission color filters (BrightLine 527/20, BrightLine 580/23, Semrock, USA) blocked residual excitation light and an imaging lens projected the fluorescence emission to the CMOS camera chip with an effective pixel size of 111 nm.

The power of the 405 nm activation laser and the respective imaging lasers were adjusted until the fluorescent molecules in the region of interest showed sufficiently fluctuating intensities for SOFI imaging. Movies for the different fluorophores were then acquired sequentially by the camera's acquisition software (HOKAWO, Hamamatsu, Japan). The Cy3-tagged structures were imaged first and further illuminated until all Cy3 fluorophores were bleached, so they would not influence the ensuing recording of the ATTO488 tagged structures. Typically, each movie consisted of 10000 frames (128 × 128 pixels) with an acquisition time of 10 ms per frame.

Data processing for fluctuation imaging was performed by the open-source Localizer software package (*Dedecker et al., 2012*). The recorded image sequences underwent second-order SOFI analyses. The resulting super-resolved images were further deconvolved employing a Richardson-Lucy algorithm with three iterations and a Gaussian point spread function (SD = 1.6 pixels).

## Co-localization analysis

Line intensity profiles of randomly selected Regions of Interest (ROI) were generated by ZEN software. Data from ZEN software were exported to Excel.

## Acknowledgments

We are grateful to Marja Timmermans and all members of the lab for critical reading of the manuscript, and to Xiuren Zhang for discussions and sharing unpublished results. This work was funded by the DFG through SFB1101 (to Y.-D.S., S. z.O.-K, M.S. and S.L.) and research grant LA2633/4-1 (to S. L.). This work was supported by the Chemical Genomics Centre (CGC) of the Max Planck Society and its supporting companies AstraZeneca, Bayer CropScience, Bayer HealthCare, Boehringer Ingelheim and Merck (to S.L).

# Additional information

## Funding

| Funder | Author |
| --- | --- |
| Deutsche Forschungsge-meinschaft | York-Dieter Stierhof Sven zur Oven-Krockhaus Sascha Laubinger |
| Max-Planck-Gesellschaft | Sascha Laubinger |

The funders had no role in study design, data collection and interpretation, or the decision to submit the work for publication.

## Author contributions

Corinna Speth, Conceptualization, Formal analysis, Investigation, Visualization, Methodology, Writing—review and editing; Emese Xochitl Szabo, Resources, Data curation, Software, Formal analysis, Investigation, Methodology, Writing—review and editing; Claudia Martinho, Formal analysis, Investigation, Visualization, Methodology, Writing—review and editing; Silvio Collani, Data curation, Formal analysis, Investigation, Methodology, Writing—review and editing; Sven zur Oven-Krockhaus, Resources, Software, Formal analysis, Investigation, Methodology, Writing—review and editing; Sandra Richter, Formal analysis, Writing—review and editing; Irina Droste-Borel, Chang Liu, Formal analysis, Investigation, Methodology, Writing—review and editing; Boris Macek, Markus Schmid, Supervision, Writing—review and editing; York-Dieter Stierhof, Resources, Formal analysis, Funding acquisition, Investigation, Methodology, Writing—review and editing; Sascha Laubinger, Conceptualization, Supervision, Funding acquisition, Writing—original draft, Project administration, Writing—review and editing

## Author ORCIDs

Silvio Collani (iD) http://orcid.org/0000-0002-9603-0882
Chang Liu (iD) http://orcid.org/0000-0003-2859-4288
Sascha Laubinger (iD) http://orcid.org/0000-0002-6682-0728

## Decision letter and Author response

Decision letter https://doi.org/10.7554/eLife.37078.035
Author response https://doi.org/10.7554/eLife.37078.036

# Additional files

## Supplementary files

• Supplementary file 1. Full list of SE binding sites in the Arabidopsis genome. The table contains all binding sites with start and end coordinates, and annotations.
DOI: https://doi.org/10.7554/eLife.37078.015

• Supplementary file 2. List of conservative SE binding site identified by RSAT. The table contains information about all significantly enriched sequences among the SE binding sites, and the distribution of the peaks among the SE binding sites.
DOI: https://doi.org/10.7554/eLife.37078.016

• Supplementary file 3. List of all oligonucleotides used in this study.
DOI: https://doi.org/10.7554/eLife.37078.017

• Supplementary file 4. List of identified proteins in mass spectrometry analysis. Immunoprecipitation experiments were conducted with antibodies against CTD-Ser2P (POL_II_S2; replicate 1–3), CTD-Ser5P (POL_II_S5; replicate 1–3), unphophorylated CTD (CTD; replicate 2–3) and the corresponding IgG controls. The table shows all proteins identified as well as the corresponding number of peptides, sequence coverage and quality measures in all biological replicates.
DOI: https://doi.org/10.7554/eLife.37078.018

• Transparent reporting form

DOI: https://doi.org/10.7554/eLife.37078.019

## Data availability

Raw data have been deposited under accession codes accession number ERP016859 (ENA), PXD006004 (Pride) and GSE99367 (Geo Omnibus).

The following datasets were generated:

| Author(s) | Year | Dataset title | Dataset URL | Database, license, and accessibility information |
|---|---|---|---|---|
| Corinna Speth, Silvio Collani, Markus Schmid, Sascha Laubinger | 2018 | SE ChIP-seq | https://www.ebi.ac.uk/ena/data/view/PRJEB15153 | Publicly available at the European Nucleotide Archive (accession no. PRJEB15153) |
| Corinna Speth, Claudia Martinho, Sascha Laubinger | 2018 | POL II IP | http://proteomecentral.proteomexchange.org/cgi/GetDataset?ID=PXD006004 | Publicly available at ProteomeXchange (accession no. PXD006004) |
| Martinho C, Speth C, Szabo EX, Laubinger S | 2018 | RNA-seq of se mutants | https://www.ncbi.nlm.nih.gov/geo/query/acc.cgi?acc=GSE99367 | Publicly available at the NCBI Gene Expression Omnibus (accession no: GSE99367) |

The following previously published datasets were used:

| Author(s) | Year | Dataset title | Dataset URL | Database, license, and accessibility information |
|---|---|---|---|---|
| Garcia EL, Matera AG, Praveen K | 2016 | Transcriptomic comparison of Drosophila snRNP biogenesis mutants reveals mutant-specific changes in pre-mRNA processing: implications for Spinal Muscular Atrophy | https://www.ncbi.nlm.nih.gov/geo/query/acc.cgi?acc=GSE81121 | Publicly available at the NCBI Gene Expression Omnibus (accession no: GSE81121) |
| Kawahara Y, Oono Y, Ogata J, Kanamori H, Sasaki H, Mori S, Matsumoto T, Itoh T | 2015 | TENOR: Database for comprehensive mRNA-seq experiments in Rice | https://trace.ddbj.nig.ac.jp/DRASearch/submission?acc=DRA000959 | Publicly available at the DDBJ Center website (accession no. DRA000959) |
| Lu X, Zhou X, Cao Y, Zhou M, McNeil D, Yang C | 2016 | RNA-Seq analysis, transcriptome assembly and gene expression profile analysis for Zea may ssp. mexicana L. under cold and drought stress | https://www.ncbi.nlm.nih.gov/geo/query/acc.cgi?acc=GSE76939 | Publicly available at the NCBI Gene Expression Omnibus (accession no: GSE76939) |
| Belamkar V, Weeks NT, Bharti AK, Farmer AD, Graham MA, Cannon SB | 2014 | Comprehensive characterization and RNA-Seq profiling of the HD-Zip transcription factor family in soybean (Glycine max) during dehydration and salt stress | https://www.ncbi.nlm.nih.gov/geo/query/acc.cgi?acc=GSE57252 | Publicly available at the NCBI Gene Expression Omnibus (accession no: GSE57252) |

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
