## [Decision Letter]

Thank you for submitting your article "*Arabidopsis* RNA processing factor SERRATE regulates the transcription of intronless genes" for consideration by *eLife*. Your article has been reviewed by three peer reviewers, and the evaluation has been overseen by a Reviewing Editor and Kevin Struhl as the Senior Editor. The reviewers have opted to remain anonymous.

The reviewers have discussed the reviews with one another and the Reviewing Editor has drafted this decision to help you prepare a revised submission.

Summary:

The *Arabidopsis* RNA binding protein SERRATE (SE) is best known for its function in primary miRNA processing. This paper reports an unexpected role for SE in promoting the transcription of a subset of intron-poor genes though promoting pol II occupancy at these genes. This paper reveals a mechanism that explains how some intron-poor genes achieve high expression in *Arabidopsis* and such mechanism might be also conserved in fly.

Essential revisions:

1) Figure 1B: provide a better illustration of exon-association with IGV browser views of selected examples.

2) Figure 2: perform additional analyses to compare SE targets and non-SE targets with same intron numbers.

3) Figure 3: the distribution patterns of pol II CTD, Ser5P, and Ser2P are very similar, please clarify and soften the statement that SE acts on paused or elongating pol II complexes if no more data can be provided to distinguish the two forms of pol II.

4) Include similar analysis (and display) of intron content in genes down-regulated in *Arabidopsis se* as in *Drosophila ars2*.

5) Provide a full list of the proteins identified by mass spectrometry in supplementary information.

---

## [Author Response]

Essential revisions:1) Figure 1B: provide a better illustration of exon-association with IGV browser views of selected examples.

We modified Figure 1B accordingly.

2) Figure 2: perform additional analyses to compare SE targets and non-SE targets with same intron numbers.

We thank the reviewers for this suggestion. We performed the analysis and the data is presented in Figure 2B and C of the revised manuscript. We found that intronless SE targets are significantly higher expressed than intronless non-SE targets. We also observed that introns could further boost the expression of SE targets.

3) Figure 3: the distribution patterns of pol II CTD, Ser5P, and Ser2P are very similar, please clarify and soften the statement that SE acts on paused or elongating pol II complexes if no more data can be provided to distinguish the two forms of pol II.

The reviewers are right. We think that the distribution patterns of pol II CTD, Ser5P, and Ser2P in our ChIP experiments are very similar because we analyzed small genes. It is much more difficult to obtain the characteristic distribution patterns because the resolution of regular ChIP experiment is not high enough for very small genes. We modified the text accordingly.

4) Include similar analysis (and display) of intron content in genes down-regulated in Arabidopsis se as in Drosophila ars2.

We are grateful for this comment. We show the results in Figure 2K of the revised manuscript. Genes expressed at lower levels in *se* mutants contained significantly less introns than *Arabidopsis* average genes.

5) Provide a full list of the proteins identified by mass spectrometry in supplementary information.

We generated an additional table (Supplementary file 4), which includes all identified proteins.